# Comparison of Different Types of Static Computer-Guided Implant Surgery in Varying Bone Inclinations

**DOI:** 10.3390/ma15093004

**Published:** 2022-04-20

**Authors:** Pisut Thangwarawut, Pokpong Amornvit, Dinesh Rokaya, Sirichai Kiattavorncharoen

**Affiliations:** 1Education Program in Implant Dentistry, Mahidol University, Bangkok 10400, Thailand; pisutthangwarawut@gmail.com; 2PPFACEDESIGN, The S Clinic, Bangkok 10120, Thailand; pokpong_am@yahoo.com; 3Department of Clinical Dentistry, Walailak University International College of Dentistry, Walailak University, Bangkok 10400, Thailand; dinesh.ro@wu.ac.th; 4Department of Oral and Maxillofacial Surgery, Mahidol University, Bangkok 10400, Thailand

**Keywords:** dental implant, accuracy, computer-guided surgery, bone inclination, surgical guide, stent

## Abstract

This research aimed to compare the accuracy of dental implant placement among three types of surgical guide: metal sleeve with key handle (Nobel guide, Nobel Biocare, Göteborg, Sweden), metal sleeve without key handle, and non-sleeve without key handle (Dentium full guide kit, Dentium Co., Seoul, Korea) when placing the implant in different bone inclinations. A total of 72 polyurethane bone models were fabricated in different inclinations (0°, 45°, and 60°). The dental implants were placed in bone models following the company’s recommendations. After dental implants were installed, the digital scans were done by an extraoral scanner. The deviations of the dental implant position were evaluated by superimposition between post-implant placement and reference model by using GOM inspect software. The deviation measurement was shown in 5 parameters: angular deviation, 3D deviation at the crest, 3D deviation at the apex, lateral linear deviation, and vertical linear deviation. The data were analyzed using One-way ANOVA and post-hoc tests at a significance level of 0.05. The accuracy of the dental implant position was not significantly influenced by the difference in the surgical guide system (*p* > 0.05). There were significant differences between placed and planned implant positions in the different inclinations of the bone. A significant difference was found in all parameters of 0° and 60° bone inclinations (*p* < 0.05). At 0° and 45°, angulated bone showed significant differences except in 3D deviation at the apex. Between 45° and 60° were significant differences only in angular deviation. Within the limitations of this study, the accuracy of implant placement among three types of surgical guides (Non-sleeve without key handle, Metal sleeve without key handle, and Metal sleeve with key handle) from two companies (Dentium and Nobel Biocare) was similar. Hence, the operators can choose the surgical guide system according to their preference. The inclination of bone can influence the angulation of dental implants.

## 1. Introduction

The dental implant is widely used in edentulous patients because of the high survival and success rates in clinical use [1,2,3,4,5]. The success of implants and implant-supported prostheses depends on the proper treatment plan. One of the most complications of implant failure is the malposition of the implant fixture leading to prosthetic failure, peri-implant failure, and unaesthetic [6]. The prosthetically driven design for positioning the dental implant fixture is a concept for the successful treatment outcome of implant-supported prostheses [7,8,9,10,11].

Implant placement can be divided into; immediate placement, early placement, and delayed, and the implant loading protocols can be classified into immediate, early, and conventional loading [12,13,14]. Some literature supports that immediate loading protocol with implant prostheses is predictable as early as conventional loading [15]. Due to the anatomy of the socket following the extraction of teeth the osteotomy drills may bend from the planned position due to the palatal bony slope of the socket resulting in a deviation of the final dental implant position [16]. Similarly, in delayed wound healing, the inclination of the bone can be found because of bone remodeling, and the placement of implant fixture should be strictly controlled in case planning the immediate restoration. Furthermore, implant placement depends on several factors, such as patient health condition, implant placement site, implant fixture, surgical procedure, and implant prosthesis-related factors [17].

Proper dental implant placement is important for the long-term success of the dental implant. The treatment planning for dental implant placement involves various aspects. Prosthetically driven positioning of the dental implant fixture has resulted in better outcomes for implant-supported prostheses. Pre-surgical radiographs help in implant planning and surgical guides/templates help in the correct placement of the dental implant [18]. Traditional 2D radiographs can provide important information about proposed implant sites but limited film size, magnification, image distortion, and 2D view restrict their use in many cases [19]., Computed tomography (CT) and cone-beam CT (CBCT) images coupled with software programs help in selecting implant dimensions and predicting treatment outcomes [19,20]. In near future, the development of artificial intelligence (AI) will assist in implant planning [21].

The dental implant placement can be done freehand and implant placement using a surgical guide. In the freehand method, the implant position is planned based on a radiograph and visual exploration of the edentulous area is described by Adell and Brånemark [22]. The osteotomy and implant fixtures are used according to the implant company’s recommendations. It is found that one of the common implant failures results from the improper position of the dental implant fixture which affects restoration, peri-implant tissue, and esthetic. Many studies revealed significantly higher errors in freehand implant placement technique than in implant placement using a surgical guide [23,24]. Many malposition implant fixtures need to be removed [25] or surgically corrected [26,27].

The accuracy of dental implant placement using the static guidance surgery system depends on many other factors. The overall deviation can occur from the plan until the placement of the fixture. Errors during data acquisition, data transfer, data processing, treatment planning, guide design, and production, as well as surgical execution, might all contribute to increased deviations from the virtually planned implant position [28,29,30,31,32,33,34,35]. Moreover, one of the most important factors that affect the accuracy of the implant placement is the patient’s condition such as age, mucosal thickness, the limitation of mouth opening, and bone density.

A surgical guide (surgical template or surgical stent) helps to guide the position and angulation of the dental implant fixture in the correct position [36]. It helps in the correct placement of the dental implant and reduces the complication of implant, such as prosthetic failure, peri-implant failure, and unaesthetic due to malposition of the implant fixture. In addition, surgical guides are useful in the anterior region where more esthetics is required and necessary to place the dental implant accurately and plan screw-retained restorations [37]. Furthermore, the surgical guide minimizes surgical complications, reduces surgical time, controls the depth of the placement to avoid damage to anatomical structures, and improves the patient’s experience [37,38].

The surgical guides can be tooth-supported, tissue-supported, and bone-supported guides which can be made from various materials such as self-cure acrylic resin, vacuum-formed polymers, metal-reinforced acrylic templates, computer-aided design, and computer-aided manufacturing (CAD-CAM) prosthesis, stereolithographic models, etc. [39,40]. The implant deviation can be influenced by the dimension of the sleeve and the component of the surgical guide which can be studied with standardized radiographs [41]. The ideal properties of a surgical guide should be rigid and stable when insert into the correct position. Other requirements of the surgical guide should be transparent because the operator can easily observe the underneath structure while drilling. The surgical guide should not be large or hard to insert into the patient’s mouth [42]. Surgical guide systems can be conventional or computer-guided systems. Furthermore, the computer-guided system can be further divided into static- and dynamic computer-assisted guidance [43]. Computer-assisted guidance surgery is more precise than conventional free-handed surgery [43,44]. However, the static approach is more commonly used than the dynamic approach because of its ease of handling and lower costs. Hence, most major implant brands have their own static computer-assisted guidance surgery system, but all are based on the same basic principle. The advantage of using static computer-assisted guidance surgery is knowing the implant fixture position before the final implant placement and helps to fabricate the immediate implant restoration on the same day [45,46,47]. Currently, the static surgical guide can be subdivided into three different systems: metal sleeve with drilling key handle type, metal sleeve without drilling key handle type, and non-sleeve without drilling key handle type.

There are several methods to evaluate the accuracy of implant placement using a surgical guide. Firstly, the implant position is directly assessed by using CBCT after implant placement. Secondly, the implant position is received from impression coping [48] or scan body [44] that is connected to the placed implant fixture.

Buser et al. [12] mentioned that the accuracy of implant placement is affected by the inclination of bone, especially in the esthetic zone which can result in esthetic complications. In addition, Gluckman et al. [16] classified the root position in the upper anterior tooth and the socket which is helpful for the planning for immediate implant placement and prosthesis fabrication for screw- and cement-retained. Although these studies show that the inclination of bone is one of the important factors for implant placement, there is no study on the accuracy of implant placement in different inclinations of bone. Hence, our study aims to study the accuracy of implant placement among three types of surgical guides from two companies (Dentium and Nobel Biocare) and to study the accuracy of implant placement in different inclinations of bone (0°, 45°, and 60° degree).

## 2. Materials and Methods

This study is an in vitro study in an experimental model system.

### 2.1. Preparation of Surgical Guides

Three types of surgical guides were prepared following 2 brand systems: Dentium full guide kit (Dentium Co., Seoul, South Korea) and Nobel guide (Nobel Biocare, Göteborg, Sweden). The surgical guide models were designed using Meshmixer 3.5 software (Autodesk, San Rafael, CA, USA), exported to STL file, and printed using the 3D printer (Form 2, Formlabs, Somerville, MA, USA) with model resin at 50-micron resolution (Figure 1).

### 2.2. Preparation of Bone Models

Seventy-two polyurethane bone models (Axson, Frankfurt am Main, Germany) simulating human bone with the cortical thickness of 1 mm and the rest of its cancellous bone were prepared. The models were made in different inclinations (0°, 45°, and 60°) at the center of the model. The distance from the upper part of the model to the bone inclination is 2 mm at both 45° and 60° (Figure 2 and Figure 3).

The guides were obtained from the printer and bone models from the mold. They were bonded together using cyanoacrylate super glue. Then the metal sleeves were inserted inside the guide in surgical guides for the metal sleeve guides (Dentium full guide kit, Nobel guide Nobel Biocare) (Figure 4).

### 2.3. Experimental Testing

Drilling was done following the protocol from each implant company as recommended. Then, the implant fixtures were inserted through the surgical guides. The implants size selected for Dentium company (Dentium superline) is 4.0 × 12 mm and for Nobel Biocare (Nobel active) is 4.3 × 13 mm.

The following groups were divided as shown in Figure 5.

Group 1 Non-sleeve without key handle (Dentium)Group 2 Metal sleeve without key handle (Dentium)Group 3 Metal sleeve with key handle (Nobel Biocare)

After placing the implant fixtures, the metal sleeves were removed, and the diameter of the holes was increased with the drill to insert the temporary abutment as the size of the temporary abutments was larger than the metal sleeve diameter.

Then, the surgical guides with the temporary abutments above the implants were scanned with the Extraoral scanner (Ceramill map 600, Amann Girrbach, Koblach, Austria) for scanning the surface. The 3D scanned data were saved in the STL file format for the deviation measurement.

### 2.4. Deviation Measurement

A standard 3D model in a cylinder shape like a temporary abutment as an implant company was designed in Meshmixer 3.5 software (Autodesk, San Rafael, CA, USA) and was printed by the 3D printer (Form 2, Formlabs, Somerville, MA, USA). Then, the standard 3D model was scanned with the Extraoral scanner (Ceramill map 600, Amann Girrbach, Koblach, Austria). The 3D scanned data were saved in the STL file format for the deviation measurement.

The experimental 3D model and standard 3D model were aligned by GOM inspect software (GOM mbH, Braunschweig, Germany) using the corners and alphabet markers of the surgical guide. At first, for each planned and placed implant *x*, *y*, and *z* coordinates were located on their long axes which were then converted into cylinders. Then, 2 points were located; the first point was the neck point of the implant (center of the most coronal portion), and the second point was the apical point of the implant (center of the implant apex) (Figure 6). Both the distance between the centers of the simulated and placed implants and the angle between the long axis of implants were measured by a single observer. The deviation measurement is using 5 parameters: Angular deviation, 3D deviation at the crest, 3D deviation at the apex, Lateral linear deviation, and Vertical linear deviation.

### 2.5. Statistical Analysis

The statistical analysis was performed using SPSS Software version 18 (SPSS Statistics; Chicago, IL, USA). To test the normality, Shapiro-Wilk Test was done at *p* value >0.05 Descriptive Statistics (Mean, SD) were used. Data entry was done in MS Excel and analyzed using a One-way analysis of variance (ANOVA) between group means. Post hoc analysis was done to compare each group.

## 3. Results

### 3.1. Results of Overall Accuracy of Implant Placement Using Surgical Guide

In this study, 72 implants placement were done by using static computer-guided implant surgery in the simulated bone models and all of them had adequate primary stability. The results of the overall implant deviation are shown in Table 1. The overall angular deviation of 1.49 ± 0.53° was seen. Maximum deviation was seen at the 3D deviation at the apex (0.85 ± 0.29 mm) and the minimum deviation was seen at the vertical linear deviation (0.19 ± 0.13 mm).

### 3.2. Results of Accuracy of Implant Placement in Three Types of the Surgical Guides

#### 3.2.1. Descriptive Statistics: Angular Deviation

The results of descriptive statistics of angular deviation in three types of surgical guides are shown in Table 2. The mean and SD of dental implant deviation in the sleeve w/ handle, non-sleeve w/o handle, and sleeve w/o handle surgical guides were 1.37 ± 0.57, 1.53 ± 0.58, and 1.57 ± 0.42, respectively. The minimum deviation was seen at the sleeve w/ handle type, although it had more instruments used for the implant placement.

#### 3.2.2. Descriptive Statistics: 3D Deviation at Coronal

The results of descriptive statistics of 3D deviation at coronal in three types of surgical guides are shown in Table 3. The mean and SD of dental implant deviation in the sleeve with handle, non-sleeve w/o handle and sleeve w/o handle surgical guides were 0.43 ± 0.22, 0.38 ± 0.15, and 0.40 ± 0.16, respectively. The minimum deviation was seen at the sleeve w/o handle type. The maximum deviation was seen at the sleeve w/ handle type.

#### 3.2.3. Descriptive Statistics: 3D Deviation at Apex

The results of descriptive statistics of 3D deviation at the apex in three types of surgical guides are shown in Table 4. The mean and SD of dental implant deviation in the sleeve w/ handle, non-sleeve w/o handle, and sleeve w/o handle surgical guides were 0.98 ± 0.30, 0.77 ± 0.28, and 0.81 ± 0.27, respectively. The minimum deviation was seen at the non-sleeve w/o handle type. The maximum deviation was found at the sleeve w/handle type.

#### 3.2.4. Descriptive Statistics: Lateral Linear Deviation

The results of descriptive statistics of lateral linear deviation in three types of surgical guides are shown in Table 5. The mean and SD of dental implant deviation in the sleeve w/ handle, non-sleeve w/o handle, and sleeve w/o handle surgical guides were 0.37 ± 0.17, 0.31 ± 0.14, and 0.36 ± 0.16, respectively. The minimum deviation was seen at the non-sleeve w/o handle type. The maximum deviation was found at the sleeve w/ handle type.

#### 3.2.5. Descriptive Statistics: Vertical Linear Deviation

The results of descriptive statistics of vertical linear deviation in three types of surgical guides are shown in Table 6. The mean and SD of dental implant deviation in the sleeve w/ handle, non-sleeve w/o handle, and sleeve w/o handle surgical guides were 0.22 ± 0.16, 0.19 ± 0.12, and 0.15 ± 0.08, respectively. The minimum deviation was seen at the sleeve w/o handle type. The maximum deviation was seen at the sleeve w/ handle type.

#### 3.2.6. Results of Multiple Comparison

The results of multiple comparisons of the three types of surgical guides for the angular deviation, 3D deviation at coronal, and 3D deviation at apex, vertical linear deviation, and lateral linear deviation for the accuracy of implant placement in three types of the surgical guides is shown in Table 7. It was found that there was no significant difference among three types of surgical guides for the Angular deviation, 3D Deviation at Coronal, and 3D deviation at apex, vertical linear deviation, and lateral linear deviation (*p* > 0.05).

### 3.3. Results of Accuracy of Implant Placement in Different Inclination of Bone

#### 3.3.1. Descriptive Statistics: Angular Deviation (degree)

The results of descriptive statistics of angular deviation in varying bone inclinations are shown in Table 8. The mean and SD of dental implant deviation in 0°, 45°, and 60° bone inclination were 1.05 ± 0.39, 1.53 ± 0.34, and 1.89 ± 0.48, respectively. The minimum deviation was seen at the 0° bone inclination. The maximum deviation was seen at 60° bone inclination.

#### 3.3.2. Descriptive Statistics: 3D Deviation at Coronal

The results of descriptive statistics of 3D deviation at coronal in varying bone inclinations are shown in Table 9. The mean and SD of dental implant deviation in 0°, 45°, and 60° bone inclination were 0.25 ± 0.10, 0.44 ± 0.14, and 0.52 ± 0.17, respectively. The minimum deviation was seen at the 0° bone inclination. The maximum deviation was seen at 60° bone inclination.

#### 3.3.3. Descriptive Statistics: 3D Deviation at Apex

The results of descriptive statistics of 3D deviation at the apex in varying bone inclinations are shown in Table 10. The mean and SD of dental implant deviation in 0°, 45°and 60° bone inclination were 0.70 ± 0.26, 0.86 ± 0.29, and 1.01 ± 0.25, respectively. The minimum deviation was seen at the 0° of bone inclination. The maximum deviation was seen at 60° bone inclination.

#### 3.3.4. Descriptive Statistics: Lateral Linear Deviation

The results of descriptive statistics of lateral linear deviation in varying bone inclinations are shown in Table 11. The mean and SD of dental implant deviation in 0°, 45°, and 60° bone inclination were 0.22 ± 0.08, 0.36 ± 0.15, and 0.45 ± 0.15, respectively. The maximum deviation was seen at 60° bone inclination.

#### 3.3.5. Descriptive Statistics: Vertical Linear Deviation

The results of descriptive statistics of vertical linear deviation in varying bone inclinations are shown in Table 12. The mean and SD of dental implant deviation in 0°, 45°, and 60° bone inclination were 0.13 ± 0.06, 0.22 ± 0.12, and 0.22 ± 0.16, respectively. The minimum deviation was seen at the 0° of bone inclination.

#### 3.3.6. Multiple Comparison

The results of multiple comparisons of the three types of surgical guides for the angular deviation, 3D deviation at coronal, and 3D deviation at apex, vertical linear deviation, and lateral linear deviation for the accuracy of implant placement in different inclination of bone are shown in Table 13. The angular deviation (degree) was significantly different among varying bone inclination (*p* < 0.05). The 3D deviation at coronal was significantly different from bone inclination (*p* < 0.05) except between 45°and 60° bone inclination. The 3D deviation at the apex was significantly different between 45°and 60° bone inclination (*p* < 0.05). The lateral linear deviation was significantly different from bone inclination (*p* < 0.05) except between 45°and 60° bone inclination. The minimum deviation was seen at the 0° of bone inclination. The vertical linear deviation was significantly different from bone inclination (*p* < 0.05) except between 45° and 60° bone inclination.

## 4. Discussion

Following the extraction of teeth, resorption of bone takes place, and the ridge is not always flat in every situation. Hence, in this study, we investigated the accuracy of implant placement using three different surgical guides from two companies (Dentium and Nobel Biocare) in different bone inclinations (0°, 45°, and 60°). This study was done considering the new factor (inclination of bone) in three systems from two implant companies (Dentium and Nobel Biocare) that may influence the dental implant position as no previous studies considered these factors. In this study, 72 implants placement were done by using static computer-guided implant surgery in the simulated bone models and all of them had adequate primary stability. We hypothesized that the accuracy of the dental implant placement is not disturbed by the bone inclinations and different surgical guide systems.

In this study, we found that the deviation of the dental implant between placed and planned implant positions was 1.49° in angulation (minimum 0.38° and maximum 3.03°). The entry point deviation was 0.40 mm (minimum 0.10 mm and maximum 0.96 mm). Our results were similar to the vitro study done by Yeung et al. [40] where they measured the accuracy of 3-D printed implant surgical guides with different implant systems and they found the angular deviation of the dental implant in mesiodistal and labiopalatal were 1.69° and 1.56°, respectively and the coronal displacement of the dental implant were 0.02 to 1.26 mm. But they did not study the apical point deviation but in our study, it was 0.85 mm (minimum 0.27 mm and maximum 1.54 mm). Similarly, a previous study by Thangwarawut et al. [49] found that the accuracy of the dental implant position was not significantly influenced by the difference in the surgical guide system. However, more deviation of the implant position was found in a metal sleeve with key handle type, except in the angular deviation.

However, our study showed lesser deviation values than the results of a systematic review and meta-analysis done by Tahmaseb et al. [50] (2238 implants in 471 patients using static computer-guided implant surgery) where they found the mean deviation of 1.2 mm at the entry point and 1.4 mm at the apical point with the angular deviation was 3.5°. The reason for less deviation of implants in our study might be due to the elimination of various factors, such as bone density, mucosal thickness, limitation of mouth opening, and the other environment in the mouth that can improve the predictability and accuracy of the implant.

In addition, our study showed that there was no significant difference among the three types of implant surgical guide systems in the same inclination of the bone condition. Hence, the operators can choose the surgical guide system according to their preference. However, it shows the trend of implant position between planned and placed in most parameters is slightly larger in the case of the surgical guide that has a metal sleeve with a handle (i.e., NobelGuide). It can be inferred that the more instruments used for the implant placement, the more will be the error. More instruments will increase the deviation of the implant position because the operator should concentrate both two hands while drilling. Moreover, the source of error was caused by discrepancies between the key handle, metal sleeve, and the drills. According to Valente et al. [51], the implant deviation is caused by the gap between the tube and the drilling burs.

Regarding the implant placement in different types of implant surgical guide systems, we found that there was a significant difference in angulation of the implant in NobelGuide and there was a significant difference in implant position in DentiumGuide. This might be because of the more component of the guide, drill bur shape, and surgical protocol. In NobelGuide, the drill burs are twisted in shape and the implant bed preparation protocol uses the same length of the drill bur from the first drill until the final drill. This causes a difference to change the implant direction/angulation if there is a mistake in the first drill. But in DentiumGuide, the drill burs (both sleeveless and metal sleeve) are taper in shape with the provision of side cutting, and the first drill bur act as a guide drill that is shorter (only 6 mm in length), slowly increasing in length till the final drill bur. Hence, the implant direction/ angulation can be corrected if there is a mistake in the first drill. But since the first drill is only 6 mm in DentiumGuide, this might cause a change in the implant position in our study.

When placing a dental implant in different bone inclinations, more implant deviation is seen in a greater degree of bone inclination. In our study, the deviation of the implant is seen in 60° in all three types of implant surgical guide systems. This might be because when the drilling bur touches the inclined surface, the bur usually slips into the soft and thin bone. Buser et al. [12] mentioned that the accuracy of implant placement is affected by the inclination of bone, especially in the esthetic zone which can result in esthetic complications. Similarly, Ozan et al. [24] studied the correlation between the dental implant deviation and the bone density of the implant sites in different types of surgical guides. They found that the lower bone density, the more angular deviations of the dental implant occur in surgical guide surgery. Likewise, less angular deviation values were found in the high-density bone.

In this study, the drill was fully guided by the surgical guide hence the drill lengths and implant position depth were correct in all types of surgical guides. In a previous study done by Yeung et al. [40] placed the implants with and without a surgical guide and found that when they did not use the surgical guide, a vertical displacement was found up to 3 mm longer than the required drill. Furthermore, the apical deviation was more than the coronal deviation in all types of surgical guides. This might be due to the presence of the part of the drill is in the sleeve of the surgical guide which caused less deviation [31]. Additionally, as the length of the drill (distance from the guide to the apex of the osteotomy) increases, the deviation also can increase.

A systematic review by Sigcho et al. [52] (2767 dental implants) found that not only showed the accuracy of the static computer-guided surgery but also demonstrated the factor that affect the accuracy of the guide. The possible causes for errors can occur in both pre-surgical planning and the surgical phase. In addition, a systematic review and meta-analysis done by Ali Tahmaseb et al. [51] (2136 implants) revealed a total mean error of 1.2 mm at the entry point, 1.4 mm at the apical point, and deviation of 3.5°. Another systematic review reported by Jung et al. [53] showed a total mean error of 0.74 mm (maximum of 4.5 mm) at the entry point and 0.85 mm (maximum of 4.5 mm) at the apex. Hence, errors are found while placing the implants. The different systems of intraoral scanners affect the accuracy of dimensions and the unsuitable distance from the scanner to the object can affect the scanner accuracy [54,55]. Moreover, the error increased with the increased area of the scanned section. The limitation of an oral scanner is that it cannot scan movable tissue well. Sometimes, free-end edentulous or completely edentulous areas may use the conventional impression with impression material for stone model and use an intraoral scanner or laboratory scanner. Then, the errors can occur in the analog to the digital procedure of a model. Implant planning software uses CT and surface scan to superimpose both two data for creating the surgical guide. Misalignment of merging files may occur from metallic restorative materials that create artifacts, or the teeth were occluded when CT resulting to unclearly see for merging.

The study in different systems of static computer-guided surgery by Laederach et al. [56] revealed the dissimilar designs of drilling with the sleeves have a significant impact on the accuracy of the implant position. Furthermore, the friction between the metal sleeves and drill handles, and the drill handles and drills can affect the implant position [31]. Schneider and colleagues [34] reported the lateral deviation decreased with a reduced diameter of a metal sleeve and found that using short drills with higher drill handles can decrease the errors. Kholy et al. [57] also revealed that reducing the drilling distance below the metal sleeve or using shorter sleeve heights can increase the accuracy of the guide.

To test the error in the deviation measurement in this study, following the completion of the study, 3 measurements in each parameter (Angular Deviation, 3D Deviation at Coronal, 3D Deviation at Apex, Lateral Linear Deviation, Vertical Linear Deviation) were randomly in 3 groups. The measurement errors were calculated by Dalhberg’s Formula [58]. The error for each parameter was below 2% which was within the acceptable limit (The error should be below 5%).

There are some limitations to this study. This study is an in vitro study and has eliminated various factors such as bone density, adjacent teeth, mucosal thickness, tongue, limitation of mouth opening, and the other environment in the mouth that can affect the accuracy of the implant. In this study, we considered only three different inclinations of bone (0°, 45°, and 60°) from 2 companies (Dentium and Nobel Biocare). Future studies can be conducted clinically to study the accuracy of implant placement in different inclinations of bone among different surgical guides from various companies.

## 5. Conclusions

The accuracy of implant placement among three types of surgical guides (Non-sleeve without key handle, Metal sleeve without key handle, and Metal sleeve with key handle) from two companies (Dentium and Nobel Biocare) was similar. Hence, the operators can choose the surgical guide system according to their preference. The inclination of bone can affect the angulation of dental implants. Therefore, the surgeon should drill and place the dental implant carefully in more inclination bone although using a fully surgical guide.

## Figures and Tables

**Figure 1 materials-15-03004-f001:**
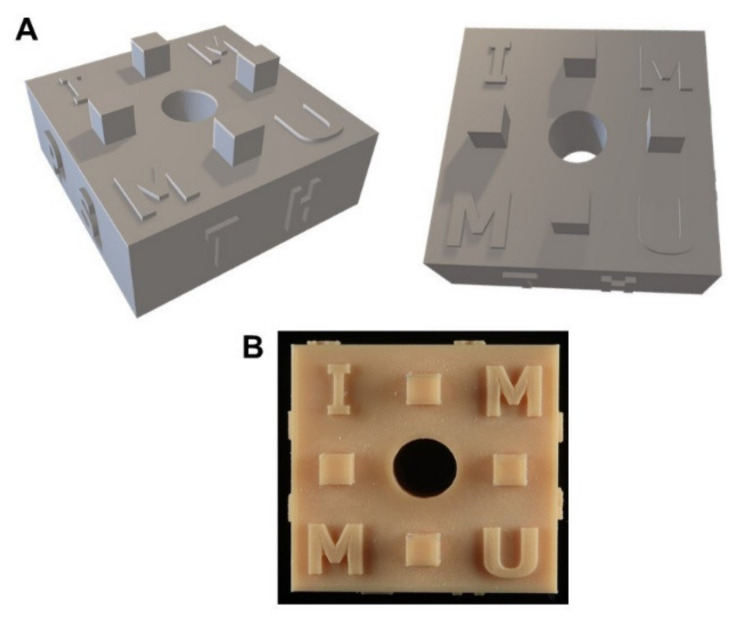
The 3D design of surgical guides (**A**) and printed surgical guide (**B**).

**Figure 2 materials-15-03004-f002:**
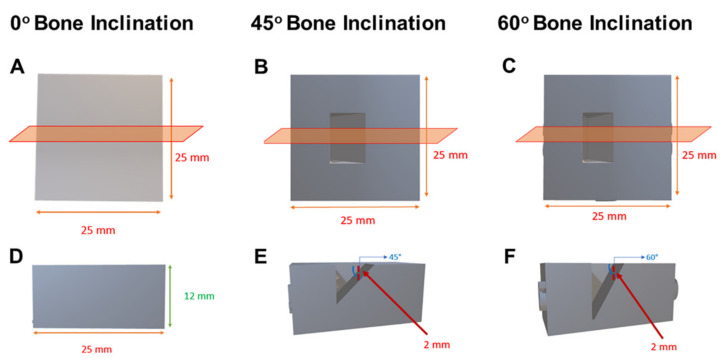
Bone models showing bone inclinations; 0° (**A**,**D**), 45° (**B**,**E**), and 60° (**C**,**F**). (**A**–**C**) are top view and (**D**–**F**) are side view.

**Figure 3 materials-15-03004-f003:**
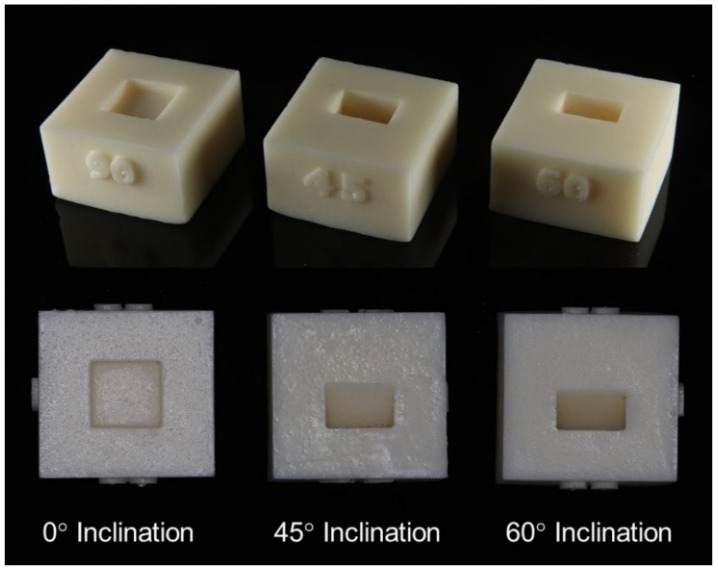
Bone models (oblique view and top view).

**Figure 4 materials-15-03004-f004:**
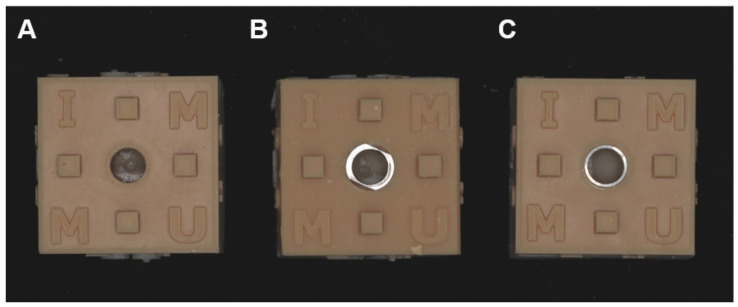
The final preparation of surgical guides with bone models. (**A**). Non-sleeve w/o key handle, (**B**). Metal sleeve w/o key handle, (**C**). Metal sleeve w/ key handle.

**Figure 5 materials-15-03004-f005:**
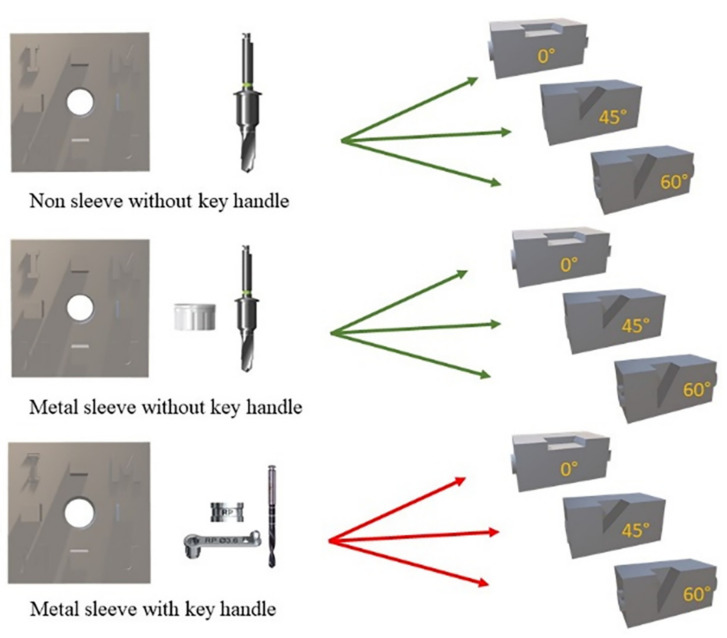
Experimental testing is divided into 3 groups.

**Figure 6 materials-15-03004-f006:**
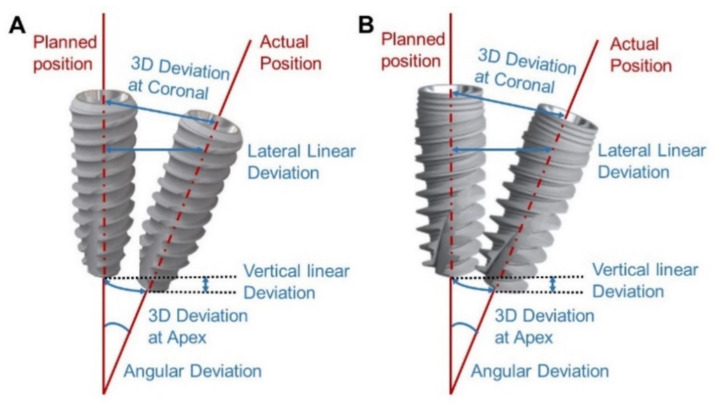
Deviation measurements and deviation parameters. (**A**). Dentium superline and (**B**). Nobel active.

**Table 1 materials-15-03004-t001:** Results of implant deviation with a variety of bone models.

Parameters	Results	Minimum	Maximum
Angular Deviation	1.49 ± 0.53°	0.38°	3.03°
3D Deviation at Coronal	0.40 ± 0.18 mm	0.10 mm	0.96 mm
3D Deviation at Apex	0.85 ± 0.29 mm	0.27 mm	1.54 mm
Lateral Linear Deviation	0.34 ± 0.16 mm	0.10 mm	0.74 mm
Vertical Linear Deviation	0.19 ± 0.13 mm	0.03 mm	0.62 mm

**Table 2 materials-15-03004-t002:** Results of descriptive statistics of angular deviation in three types of surgical guides.

Types	*n*	Mean (Degree)	SD	Minimum	Maximum
Sleeve w/handle	24	1.37	0.57	0.38	2.79
Non-Sleeve w/o handle	24	1.53	0.58	0.64	3.03
Sleeve w/o handle	24	1.57	0.42	0.59	2.29
Total	72	1.49	0.53	0.38	3.03

SD = standard deviation, w/ = with, and w/o = without.

**Table 3 materials-15-03004-t003:** Results of descriptive statistics of 3D deviation at coronal in three types of surgical guides.

Types	*n*	Mean (mm)	SD	Minimum	Maximum
Sleeve w/handle	24	0.43	0.22	0.10	0.96
Non-Sleeve w/o handle	24	0.38	0.15	0.15	0.70
Sleeve w/o handle	24	0.40	0.16	0.12	0.71
Total	72	0.40	0.18	0.10	0.96

SD = standard deviation, w/ = with, and w/o = without.

**Table 4 materials-15-03004-t004:** Results of descriptive statistics of 3D deviation at the apex in three types of surgical guides.

Types	*n*	Mean (mm)	SD	Minimum	Maximum
Sleeve w/ handle	24	0.98	0.30	0.40	1.54
Non-Sleeve w/o handle	24	0.77	0.28	0.30	1.24
Sleeve w/o handle	24	0.81	0.27	0.27	1.34
Total	72	0.85	0.30	0.27	1.54

SD = standard deviation, w/ = with, and w/o = without.

**Table 5 materials-15-03004-t005:** Results of descriptive statistics of lateral linear deviation in three types of surgical guides.

Types	*n*	Mean (mm)	SD	Minimum	Maximum
Sleeve w/handle	24	0.37	0.17	0.16	0.74
Non-Sleeve w/o handle	24	0.31	0.14	0.11	0.54
Sleeve w/o handle	24	0.36	0.16	0.10	0.68
Total	72	0.34	0.16	0.10	0.74

SD = standard deviation, w/ = with, and w/o = without.

**Table 6 materials-15-03004-t006:** Results of descriptive statistics of vertical linear deviation in three types of surgical guides.

Types	*n*	Mean (mm)	SD	Minimum	Maximum
Sleeve w/ handle	24	0.22	0.16	0.03	0.62
Non-Sleeve w/o handle	24	0.19	0.12	0.04	0.46
Sleeve w/o handle	24	0.15	0.08	0.03	0.29
Total	72	0.19	0.12	0.03	0.62

SD = standard deviation, w/ = with, and w/o = without.

**Table 7 materials-15-03004-t007:** The multiple comparisons in the accuracy of implant placement in three types of surgical guides.

	*p* Value		
	Sleeve w/ Handle vs. Non-Sleeve w/o Handle	Sleeve w/ Handle vs. Sleeve w/o Handle	Non-Sleeve w/o Handle vs. Sleeve w/o Handle
Angular deviation	0.596	0.453	0.970
3D deviation at coronal	0.581	0.812	0.923
3D deviation at apex	0.052	0.137	0.90
Lateral linear deviation	0.392	0.963	0.547
Vertical linear deviation	0.624	0.141	0.589

w/ = with, and w/o = without. Significant difference at *p* < 0.05. Multiple comparisons were done using One-way ANOVA.

**Table 8 materials-15-03004-t008:** Results of descriptive statistics of angular deviation in varying bone inclinations.

Bone Inclination	*n*	Mean (Degree)	SD	Minimum	Maximum
0°	24	1.05	0.39	0.38	1.92
45°	24	1.53	0.34	0.69	2.09
60°	24	1.89	0.48	0.96	3.03
Total	72	1.49	0.53	0.38	3.03

SD = standard deviation.

**Table 9 materials-15-03004-t009:** Results of descriptive statistics of 3D deviation at coronal in varying bone inclinations.

Bone Inclination	*n*	Mean (mm)	SD	Minimum	Maximum
0°	24	0.25	0.10	0.10	0.40
45°	24	0.44	0.14	0.23	0.75
60°	24	0.52	0.17	0.19	0.96
Total	72	0.40	0.18	0.10	0.96

SD = standard deviation.

**Table 10 materials-15-03004-t010:** Results of descriptive statistics of 3D deviation at apex in varying bone inclinations.

Bone Inclination	*n*	Mean (mm)	SD	Minimum	Maximum
0°	24	0.70	0.26	0.27	1.22
45°	24	0.86	0.29	0.30	1.29
60°	24	1.01	0.25	0.55	1.54
Total	72	0.85	0.30	0.27	1.54

SD = standard deviation.

**Table 11 materials-15-03004-t011:** Results of descriptive statistics of lateral linear deviation in varying bone inclinations.

Bone Inclination	*n*	Mean (mm)	SD	Minimum	Maximum
0°	24	0.22	0.08	0.10	0.38
45°	24	0.36	0.15	0.16	0.66
60°	24	0.45	0.15	0.18	0.74
Total	72	0.34	0.16	0.10	0.74

SD = standard deviation.

**Table 12 materials-15-03004-t012:** Results of descriptive statistics of vertical linear deviation in varying bone inclinations.

Bone Inclination	*n*	Mean (mm)	SD	Minimum	Maximum
0°	24	0.13	0.06	0.06	0.26
45°	24	0.22	0.12	0.03	0.51
60°	24	0.22	0.16	0.03	0.62
Total	72	0.19	0.13	0.03	0.62

SD = standard deviation.

**Table 13 materials-15-03004-t013:** The multiple comparisons of the accuracy of implant placement in varying bone inclinations.

	*p* Value		
	0 vs. 45 Degree	0 vs. 60 Degree	45 vs. 60 Degree
Angular deviation	0.001 *	<0.0001 *	0.014 *
3D deviation at coronal	<0.0001 *	<0.0001 *	0.131
3D deviation at apex	0.124	0.001 *	0.136
Lateral linear deviation	0.004 *	<0.0001 *	0.083
Vertical linear deviation	0.040 *	0.036 *	0.999

* Significant difference at *p* < 0.05. Multiple comparisons were done using One-way ANOVA.

## Data Availability

The data used to support the findings of this study are available from the corresponding author upon reasonable request.

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
