# Peer review of "Comparison of Different Types of Static Computer-Guided Implant Surgery in Varying Bone Inclinations"

_materials, 2022, doi:10.3390/ma15093004_

Round 1

Reviewer 1 Report

The study is relatively a basic research as the test model does not simulate human jaw or oral cavity in any sense at all. It is OK but its clinical relevance can thus be quite limited.

Regarding the measurements, a single observer did the measurements once, if I understand correctly. Perhaps the observer should measure again, to report the intra-observer agreement. Or another observer can measure, to report the inter-observer agreement. Either one should be fine. This is to support that the measurements are reliable. 

Author Response

The study is relatively a basic research as the test model does not simulate human jaw or oral cavity in any sense at all. It is OK but its clinical relevance can thus be quite limited.

Thank you for your positive comments. All corrections in the Manuscript for Reviewer 1 are highlighted in Yellow color.

Point 1: Regarding the measurements, a single observer did the measurements once, if I understand correctly. Perhaps the observer should measure again, to report the intra-observer agreement. Or another observer can measure, to report the inter-observer agreement. Either one should be fine. This is to support that the measurements are reliable. 

Response: All the measurements were done by one researcher. To test the error in the deviation measurement in this study, following the completion of the study, 3 measurements in each parameter (Angular Deviation, 3D Deviation at Coronal, 3D Deviation at Apex, Lateral Linear Deviation, Vertical Linear Deviation) were randomly in 3 groups. The measurement errors were calculated by Dalhberg’s Formula. The error for each parameter was below 2% which was within the acceptable limit (The error should be below 5%). (Added in the Discussion. Line 465-470)

Reviewer 2 Report

Dear authors,

the article is interesting, but it needs some improvements for publication.

The paper is a case-control study?

Please define the type of the article in the material and methods.

In the introduction please go in deeper to explain the concept of the implant planning and the methods that you use to evaluate radiography and to plan the surgical implant placement.

The factors that can influence the implant deviation such as the dimension of the sleeve and the component of the surgical guide, could be studied with stardardized radiographs. In this case please cite in the introduction the following article.

Cosola S, Toti P, Peñarrocha-Diago M, Covani U, Brevi BC, Peñarrocha-Oltra D. Standardization of three-dimensional pose of cylindrical implants from intraoral radiographs: a preliminary study. BMC Oral Health. 2021 Mar 6;21(1):100. doi: 10.1186/s12903-021-01448-9. 

In the results you should explain better and the figure must be without the company logo, you should just add the company name and reference.

In the discussion you should mention the limitation of the study.

Author Response

Dear authors, the article is interesting, but it needs some improvements for publication.

Response: Corrections in the Manuscript are highlighted in Green color.

The paper is a case-control study? Please define the type of the article in the material and methods.

Response: This study is an in vitro study in the experimental model system. (Added in the Materials and Method, Line 134)

In the introduction please go in deeper to explain the concept of the implant planning and the methods that you use to evaluate radiography and to plan the surgical implant placement.

Response: In the introduction, we have explained the concept of implant planning and the methods that we use to evaluate radiography and plan the surgical implant placement. (Line 58-68)

The factors that can influence the implant deviation such as the dimension of the sleeve and the component of the surgical guide, could be studied with stardardized radiographs. In this case please cite in the introduction the following article. Cosola S, Toti P, Peñarrocha-Diago M, Covani U, Brevi BC, Peñarrocha-Oltra D. Standardization of three-dimensional pose of cylindrical implants from intraoral radiographs: a preliminary study. BMC Oral Health. 2021 Mar 6;21(1):100. doi: 10.1186/s12903-021-01448-9.

Response: The article is cited in the introduction. (Line 100-102)

In the results you should explain better and the figure must be without the company logo, you should just add the company name and reference.

Response: The company names are removed from Figure 5.

In the discussion you should mention the limitation of the study.

Response: Limitations in this study are added at the end of the discussion. (Line 471-475)

Reviewer 3 Report

This research aimed to compare the accuracy of dental implant placement among three types of surgical guide and  three different bone inclinations.The design of the study is appropriate.

Introduction provides adequate information on literature data. Materials and methods are properly described. However results should be written in a more systematic way, with graphic representations.

Discussions  should be limited to the present study, the article is not a review. 

For conclusions it is advisable to contain more exact data regarding the parameters taken into the study.

Author Response

This research aimed to compare the accuracy of dental implant placement among three types of surgical guide and  three different bone inclinations.The design of the study is appropriate.

Thank you. All corrections in the Manuscript are highlighted in Turquoise.

Introduction provides adequate information on literature data. Materials and methods are properly described. However results should be written in a more systematic way, with graphic representations.

Response: The presentation of the results is improved.

Discussions  should be limited to the present study, the article is not a review. 

Response: Discussion is edited, shortened, and improved.

For conclusions it is advisable to contain more exact data regarding the parameters taken into the study.

Response: The conclusion is added and improved.